# A Mathematical Model to Estimate Chemotherapy Concentration at the Tumor-Site and Predict Therapy Response in Colorectal Cancer Patients with Liver Metastases

**DOI:** 10.3390/cancers13030444

**Published:** 2021-01-25

**Authors:** Daniel A. Anaya, Prashant Dogra, Zhihui Wang, Mintallah Haider, Jasmina Ehab, Daniel K. Jeong, Masoumeh Ghayouri, Gregory Y. Lauwers, Kerry Thomas, Richard Kim, Joseph D. Butner, Sara Nizzero, Javier Ruiz Ramírez, Marija Plodinec, Richard L. Sidman, Webster K. Cavenee, Renata Pasqualini, Wadih Arap, Jason B. Fleming, Vittorio Cristini

**Affiliations:** 1Department of Gastrointestinal Oncology, H. Lee Moffitt Cancer Center & Research Institute, Tampa, FL 33612, USA; mintallah.haider@moffitt.org (M.H.); jehab@health.usf.edu (J.E.); richard.kim@moffitt.org (R.K.); Jason.Fleming@moffitt.org (J.B.F.); 2Mathematics in Medicine Program, Houston Methodist Research Institute, Houston, TX 77030, USA; pdogra@houstonmethodist.org (P.D.); zwang@houstonmethodist.org (Z.W.); jbutner@houstonmethodist.org (J.D.B.); snizzero@houstonmethodist.org (S.N.); javier@g.clemson.edu (J.R.R.); 3Department of Diagnostic Imaging and Interventional Radiology, H. Lee Moffitt Cancer Center & Research Institute, Tampa, FL 33612, USA; Daniel.jeong@moffitt.org (D.K.J.); Masoumeh.Ghayouri@moffitt.org (M.G.); Gregory.Lauwers@moffitt.org (G.Y.L.); Kerry.Thomas@moffitt.org (K.T.); 4Biozentrum and the Swiss Nanoscience Institute & ARTIDIS AG, University of Basel, 4056 Basel, Switzerland; marija.plodinec@artidis.com; 5Department of Neurology, Harvard Medical School, Boston, MA 02115, USA; richard_sidman@hms.harvard.edu; 6Ludwig Institute for Cancer Research, University of California-San Diego, La Jolla, CA 92093, USA; wcavenee@ucsd.edu; 7Rutgers Cancer Institute of New Jersey & Division of Cancer Biology, Department of Radiation Oncology, Rutgers New Jersey Medical School, Newark, NJ 07103, USA; rp946@newark.rutgers.edu; 8Rutgers Cancer Institute of New Jersey & Division of Hematology/Oncology, Department of Medicine Rutgers New Jersey Medical School, Newark, NJ 07103, USA; wa116@newark.rutgers.edu

**Keywords:** chemotherapy, colorectal cancer, FOLFOX, liver metastases, mathematical model

## Abstract

**Simple Summary:**

It is known that drug transport barriers in the tumor determine drug concentration at the tumor site, causing disparity from the systemic (plasma) drug concentration. However, current clinical standard of care still bases dosage and treatment optimization on the systemic concentration of drugs. Here, we present a *proof of concept* observational cohort study to accurately estimate drug concentration at the tumor site from mathematical modeling using biologic, clinical, and imaging/perfusion data, and correlate it with outcome in colorectal cancer liver metastases. We demonstrate that drug concentration at the tumor site, not in systemic circulation, can be used as a credible biomarker for predicting chemotherapy outcome, and thus our mathematical modeling approach can be applied prospectively in the clinic to personalize treatment design to optimize outcome.

**Abstract:**

Chemotherapy remains a primary treatment for metastatic cancer, with tumor response being the benchmark outcome marker. However, therapeutic response in cancer is unpredictable due to heterogeneity in drug delivery from systemic circulation to solid tumors. In this proof-of-concept study, we evaluated chemotherapy concentration at the tumor-site and its association with therapy response by applying a mathematical model. By using pre-treatment imaging, clinical and biologic variables, and chemotherapy regimen to inform the model, we estimated tumor-site chemotherapy concentration in patients with colorectal cancer liver metastases, who received treatment prior to surgical hepatic resection with curative-intent. The differential response to therapy in resected specimens, measured with the gold-standard Tumor Regression Grade (TRG; from 1, complete response to 5, no response) was examined, relative to the model predicted systemic and tumor-site chemotherapy concentrations. We found that the average calculated plasma concentration of the cytotoxic drug was essentially equivalent across patients exhibiting different TRGs, while the estimated tumor-site chemotherapeutic concentration (eTSCC) showed a quadratic decline from TRG = 1 to TRG = 5 (*p* < 0.001). The eTSCC was significantly lower than the observed plasma concentration and dropped by a factor of ~5 between patients with complete response (TRG = 1) and those with no response (TRG = 5), while the plasma concentration remained stable across TRG groups. TRG variations were driven and predicted by differences in tumor perfusion and eTSCC. If confirmed in carefully planned prospective studies, these findings will form the basis of a paradigm shift in the care of patients with potentially curable colorectal cancer and liver metastases.

## 1. Introduction

Systemic cytotoxic chemotherapy is still the primary treatment for many cancer patients with solid tumors who present with de novo metastatic disease. Specifically, for patients with high-risk potentially curable tumors and good performance status, upfront or neoadjuvant (prior to tumor resection) chemotherapy is often the recommended approach. Response to therapy has consistently been shown to be a critical determinant of prognosis and outcomes for most systemically administered treatments, including chemotherapy drugs, biologic targeted agents, cell and/or antibody-based immunotherapy [1,2,3,4,5,6,7]. However, there is substantial heterogeneity in the overall response to treatment within and across cancer types, suggesting a puzzling differential chemotherapy effect even among patients with apparently similar clinical features of the disease [8,9,10,11]. Unfortunately, such variability in response to chemotherapy in patients remains poorly understood.

Previous works have reported on the heterogeneity of chemotherapy delivery to tumor cells as a plausible explanation for the observed differences in response to chemotherapy. Koay et al. showed that differences in inter-patient [12], as well as intra-patient [13] transport properties of pancreatic ductal adenocarcinoma tumors, lead to variation in DNA incorporation of gemcitabine, despite similar vascular pharmacokinetics of the drug [12]. Batchelor et al. highlighted the importance of improved tumor drug delivery through enhanced tumor perfusion, in order to improve patient survival in newly diagnosed glioblastoma [14]. Further, the role of improved tumor blood vessel functionality in determining the success of metronomic cancer chemotherapy was highlighted by the same group of investigators lead by Jain [15]. The above studies indicate that poor transportability of drugs through the tumor may be a driving cause of therapy failure.

Mathematical modeling efforts to understand cancer biology [16,17] and tumor growth dynamics [18,19,20] have provided support to improve therapeutic responses through conventional [21,22,23] or novel treatment strategies [24,25,26,27,28,29,30,31,32], and to overcome drug resistance [33]. Current clinical practice of delivering chemotherapy aims to achieve a systemic chemotherapy concentration ($\sigma_{B}$), based on patient-specific body measurements (e.g., body surface area (BSA)). However, inherent biophysical barriers of the tumor tissue might perhaps prevent the attainment of chemotherapy concentration at the tumor-site ($\langle \sigma_{T}\rangle$) that is necessary and sufficient to cause tumor cell death (characterized as $f_{\mathrm{kill}}$ or the fractional tumor killed by chemotherapy). Thus, tumor-specific parameters, in addition to patient characteristics, should be considered during dose design. Drug concentration at the tumor site can differ significantly from systemic concentrations, and may be a better determinant of therapeutic efficacy. To correctly estimate drug concentration at the tumor site, we have developed and validated a mechanistic mathematical model of diffusion-based drug transport, based on patient-derived data, to account for and describe the role of tumor biophysical attributes in drug efficacy [34,35,36,37]. Our mathematical model allows for the estimation of drug concentration at the tumor-site based on systemic drug pharmacokinetics, patient features, and tumor characteristics quantified through routine clinical parameters and standard imaging. Our working hypothesis is that impaired drug transport to and within the tumor microenvironment is correlated in a causal, mechanistic fashion, to drug efficacy. We first introduced this hypothesis in a histopathology-based study of response to chemotherapy [34], where we developed the mechanistic model to predict tumor response based on patient-specific measurements of tumor vascularization and perfusion. To adapt the model for determination of estimated tumor-site chemotherapeutic concentration (termed eTSCC), we designed a proof-of-concept study implementing the mechanistic perfusion model based on the tumor and treatment data of cancer patients, and examined the association between chemotherapy concentration at the tumor-site and response to therapy. Notably, we employed a colorectal liver metastasis (CRLM) model in which tumor specimens are obtained during curative-intent surgery after patients received neoadjuvant systemic cytotoxic chemotherapy. This clinical scenario provided a unique opportunity to test our mathematical model against the current gold-standard of pathologic tumor response to therapy (Tumor Regression Grade (termed TRG)) [38]. As a consequence, we were able to understand the eTSCC dependent heterogeneity in tumor response and to establish the eTSCC threshold for the classification of patients into responders and non-responders. To the best of our knowledge, this is the first model that utilizes routinely measured clinical parameters as surrogates for model parameters to estimate local chemotherapeutic drug concentration at the tumor-site. In the long term, the prospective application of this model in the clinic will enable patient-specific treatment protocol design using routine clinical measurements of patient and tumor micronevironmental parameters.

## 2. Results

On the basis of chemotherapy regimens given to the patients and tumor attributes, we have implemented a validated mathematical model to estimate chemotherapy concentration at the tumor-site and to examine its association with response to chemotherapy, measured with the current pathological gold-standard, TRG. A serial cohort of patients (*n* = 33) presenting with colorectal cancer and liver metastases (from 2016 to 2018), which received standard neoadjuvant (preoperative) chemotherapy prior to liver resection and with all the required information was included in the analysis (Table 1). As described in the Section 4, we have set out to incorporate the following parameters in the mathematical model: (i) the specific chemotherapy regimen received by all individual patients (including BSA-based dose received, number of treatment cycles, and frequency of cycles), (ii) the tumor and liver perfusion assessed through standard computed tomography image analysis [12,13,39], and (iii) the total tumor burden. We have calculated chemotherapy plasma and tumor-site concentrations (blinded to the TRG results) and subsequently used them to examine differences in response to therapy (Figure 1).

Mathematical formulation for estimating tumor-site chemotherapy drug concentration and correlation analysis. With established pharmacokinetic (PK) models of systemic drug disposition and validated models of diffusion-based drug transport, we developed a mathematical model with patient and tumor-derived clinical data to estimate the chemotherapy concentration at the tumor-site. As described in the Section 4, we first calculated the time-averaged systemic blood concentration (${\overset{=}{\sigma }}_{B,i}$) of the cytotoxic drug 5-fluorouracil (5-FU) during a chemotherapy cycle i with a standard two-compartment PK model (Equations (6) and (7)), which includes literature-derived PK parameters such as clearance and inter-compartment mass transfer rate constants [40,41,42], and patient-specific BSA-based dose (Appendix A). From ${\overset{=}{\sigma }}_{B,i}$, drug concentration in the tumor extravascular tissue ($\langle \sigma_{T}\rangle$) was then estimated with the mathematical model. As detailed in the Section 4, under the typical tumor vascularization conditions for CRLM (and other hypovascular tumors), one can safely assume that the blood volume fraction (BVF), which represents tumor perfusion, ≪1, thereby leads to the simplified form of our mathematical model:(1)$$f_{\mathrm{kill}}\sim f_{\mathrm{kill}}^{0}\cdot \mathrm{BVF}$$
where $f_{\mathrm{kill}}^{0}$ is the fraction of tumor death or cell kill (dimensionless: $0\leq f_{\mathrm{kill}}^{0}\leq 1$) that would be obtained in a dose-response study in vitro, for the given drug-cell pair at a concentration equal to the time-averaged concentration achieved in the tumor blood vessels ($\langle \sigma_{T,B}\rangle$), throughout the total duration t of the treatment. Clinically, and supported by correlation analysis of tumor and treatment variables (Figure 2A–D), an accurate surrogate for the in vitro $f_{\mathrm{kill}}^{0}$ is thus obtained by estimating $\langle \sigma_{T,B}\rangle$ as:(2)$$f_{\mathrm{kill}}^{0}\sim \langle \sigma_{T,B}\rangle \sim {\overset{=}{\sigma }}_{B,i}\cdot \frac{\lambda \cdot \Delta t}{\mathrm{CEA}}$$
where $\lambda =\frac{N}{t}$ is the frequency of chemotherapy cycles, i.e., number of cycles N per unit duration t of treatment; Δt is the duration of a single therapy cycle (≈46 h); and serum carcinoembryonic antigen (CEA) is the biomarker for total tumor burden, a surrogate for tumor size based on the correlation (Figure 2A). The significant correlation of the $\frac{\lambda }{\mathrm{CEA}}$ factor with TRG further supports and characterizes this variable in the model (Figure 2B). As described in the Section 4, we have recently reported a correlation between the BVF, measured from histopathology, and the tumor perfusion measured by a standardized approach derived from CT imaging of the tumor [34]. Our current data confirm the correlation of perfusion with a response to therapy (Figure 2C), as well as the inverse linear correlation of the size of the tumor with perfusion (Figure 2D). Thus, tumor perfusion normalized to healthy liver perfusion values serve as a surrogate for the BVF. We have previously shown that the cumulative uptake of drug over time by tumor cells in vitro, over the course of drug exposure, is a key determinant of $f_{\mathrm{kill}}$ [43]. Therefore, the time-averaged concentration $\langle \sigma_{T}\rangle$ experienced by the tumor cells throughout the course of treatment is proportional to $f_{\mathrm{kill}}$, and was obtained from the model as:(3)$$\langle \sigma_{T}\rangle \sim f_{\mathrm{kill}}^{0}\cdot \mathrm{BVF}$$

Estimated tumor-site chemotherapy concentration and correlation with tumor and treatment variables. Based on the estimated tumor-site concentration $\langle \sigma_{T}\rangle$ of 5-Fluorouracil (5-FU) (estimated from Equation (3)), we next identified significant correlations between $\langle \sigma_{T}\rangle$ and various tumor and treatment-related variables. Figure 3A illustrates the linear correlation between $\langle \sigma_{T}\rangle$ and tumor perfusion. Furthermore, as seen in Figure 3B, $\langle \sigma_{T}\rangle$ has a non-linear (power law) dependence on tumor size (cm), which can be explained by the negative linear relation observed between perfusion and tumor size in Figure 2D, leading to reduced drug accumulation in large tumors. As mentioned before, we observed a direct linear correlation between tumor size and serum CEA levels (Figure 2A); as a result, $\langle \sigma_{T}\rangle$ has a similar power law correlation with CEA (Figure 3C), as seen between $\langle \sigma_{T}\rangle$ and size (with comparable exponent values: 0.94 and 1.02). This provides a rationale for the use of serum CEA levels as a surrogate for the total tumor burden. Finally, the linear correlations between $\frac{\lambda }{\mathrm{CEA}}$ and $\langle \sigma_{T}\rangle$ (Figure 3D) and between $\frac{\lambda }{\mathrm{CEA}}$ and TRG (Figure 2B) indicate the dependence of tumor response (TRG) on $\langle \sigma_{T}\rangle$, such that a greater $\frac{\lambda }{\mathrm{CEA}}$ value corresponds to higher $\langle \sigma_{T}\rangle$ and a better response.

*Estimated chemotherapy drug concentrations and response to therapy.* Having calculated chemotherapy concentration in the systemic circulation (with the PK model) and estimated tumor-site chemotherapy concentration (with the above mathematical model (Equation (3)), we next focused on the clinical relevance of these observations by comparing the results at the two sites and the association with response to chemotherapy, i.e., TRG (Figure 4). We found that the average plasma concentration of 5-FU (${\overset{=}{\sigma }}_{B,i}$, red squares) during a therapy cycle is essentially equivalent (i.e., minimal variability) across patients exhibiting different TRGs (one-way ANOVA, *p* > 0.05), while $\langle \sigma_{T}\rangle$ shows a significant quadratic decline (one-way ANOVA, *p* < 0.001) as we move from TRG = 1 to TRG = 5 (blue circles). As a result, for instance, the average $\langle \sigma_{T}\rangle$ for patients exhibiting TRG = 5 is ~5.7 times less than their corresponding average ${\overset{=}{\sigma }}_{B,i}$. Based on these empirical observations, a number of critical findings can be summarized as follows: the chemotherapy concentration at the tumor-site (Figure 4, blue) is significantly lower than the observed plasma concentration (Figure 4, red), and drops significantly between patients with complete response (TRG = 1) and those with no response (TRG = 5), by a factor of ~4.8, while the plasma concentration (Figure 4, red) remains stable across TRG groups. Notably, the plasma concentration of 5-FU is higher than its average in vitro IC_50_ value for the same tumor type (Figure 4, black) [42], whereas the tumor-site concentration gradually falls below the mean IC_50_ for increasing TRG (i.e., poor response).

Predictive model for response to chemotherapy. To further validate the hypothesis that eTSCC ($\langle \sigma_{T}\rangle$) varied between responders and non-responders, we performed logistic regression-based binary classification analysis to classify patients based on their eTSCC. The obtained ROC curve had an AUC or *c*-statistic of ~0.88, which indicates good classification ability of eTSCC (Figure 5A). From the ROC curve, 0.51 µg mL^−1^ was selected as the optimal cut-off value to differentiate the patients into responders and non-responders (values > 0.51 µg mL^−1^ indicate responders, while <0.51 µg mL^−1^ indicate non-responders). To visualize the binary classification based on the chosen threshold, we plotted the complementary cumulative distribution (CCD) function of the eTSCC data. As shown in Figure 5B, the eTSCC correctly classified ~83% and ~85% of responders and non-responders, respectively, with an overall accuracy of classification being ~85%.

To evaluate the predictive ability of the binary classifier, we performed leave-one-out cross validation (LOOCV). The average AUC thus obtained for the ROC curves generated by iteratively removing one data point from the training data was 0.87 ± 0.01, which was very similar to the AUC of the complete training data set (Figure 5C). Based on each training data, we classified the left-out test data point, pooled the results of all the iterations, and obtained a classification accuracy of 81.8%, with ~83.3% and ~81.5% of responders and non- responders being correctly classified, respectively (Figure 5D).

## 3. Discussion

With a validated tissue perfusion model and incorporating relevant treatment and tumor-related variables, here we report the development of a mathematical model to estimate chemotherapy drug concentration at the tumor-site, and its association with response to treatment. We examined the eTSCC and response to therapy in a cohort of patients with colorectal cancer liver metastases treated with standard preoperative chemotherapy regimens prior to curative-intent surgery. This allowed us to use relatively uniform patient data from real-life clinical practices to develop and implement our models, and to measure response to chemotherapy with the current histopathologic gold-standard: TRG score (measured in surgical specimens). Our findings emphasize principles regarding drug delivery, drug penetration, and actual chemotherapy concentration achieved at the tumor site. With established physical laws of a diffusion-based validated mechanistic model to estimate chemotherapy concentration at the tumor site, we found eTSCC to be significantly lower than the calculated plasma concentration across non-responders. Most notably, the eTSCC, which is a function of tumor burden, tumor perfusion, and cumulative chemotherapy dose overtime, was directly correlated with response to chemotherapy. Moreover, this model can reliably estimate tumor site concentration, which in turn accurately predicts and classifies patients based on (expected) the response to chemotherapy. This is a novel and unique tool with potentially direct and immediate implications on day-to-day cancer care practices, as it can guide the use of chemotherapy and can help develop and implement ways to optimize the delivery of chemotherapy agents to the tumor site.

Based on the report of our previously validated hypothesis [34], which describes the direct association of tumor perfusion and response to therapy, we first focused on implementing the model to estimate chemotherapy concentration in the tumor vasculature. Though a systemic (plasma) concentration (${\overset{=}{\sigma }}_{B,i}$) is predictable with established PK models [40,41], delivery of a chemotherapy agent to the tumor site, diffusion across the microenvironment, and concentration accomplished at the site of cancerous cells ($\langle \sigma_{T}\rangle$) are more complex and to some extent erratic and difficult to measure [44]. Different techniques using in vitro and in vivo models have been developed to examine the delivery of chemotherapy agents [45,46,47], and though important for understanding the tumor delivery of agents, the complexity of this system makes these measurements challenging and far from practical for clinical use. Our findings, entered into a mathematical model, account for tumor vascularization and biophysical barriers of diffusion, such as increased interstitial fluid pressure and dense extracellular matrix, which ultimately determine the transport of the drug molecules to the tumor site. With established measures of normal tissue and tumor tissue perfusion [12,13,39], and empirical correlations of tumor burden and treatment variables, all derived from data available during the process of clinical care, we were able to estimate the tumor site chemotherapy concentration for the patients, and predict a significantly lower and variable concentration to that observed in the systemic circulation. The model developed incorporates objective data, providing a reliable measure for the estimation of tumor site concentration. Notably, results from the correlation analysis emphasized the linear association of tumor perfusion and eTSCC; this finding further supports the critical role of tumor tissue perfusion as an adequate surrogate for microenvironmental diffusion barriers. Tumor tissue diffusion barriers are of key importance for the prediction of clinical therapeutic outcomes, particularly when treating patients with hypovascular (i.e., poorly perfused) tumors, as in the well established case patients with CRLM [48].

Therefore, among the most salient findings of our study is the direct association between eTSCC and response to chemotherapy treatment. Most commonly, studies evaluating the response to treatment have focused on cellular and genetic mechanisms driving such responses [33,44,49,50]. However, our group and others have reported on the critical role of chemotherapy agent delivery, tumor perfusion, and chemotherapy penetration to reach tumor cells in governing the response to therapy [33,34,51,52,53]. After grading the response to chemotherapy with the TRG score in a blinded fashion, we found an inverse correlation between eTSCC, derived from the developed model, and the TRG score, whereas the systemic concentration remained stable and above the mean IC_50_ across all patients. This difference is based on limitations in drug delivery associated with tumor perfusion impairment, tumor burden (serum CEA level was used as a surrogate), and frequency of treatment (cumulative chemotherapy given over time). Further, eTSCC was correlated with response to treatment, and none of the traditional clinical predictive factors were found to impact this association (Appendix A). As a proof of concept study, the implications of this finding are substantial and medically meaningful as a response to therapy is the most important prognostic factor in patients with advanced solid malignancies receiving chemotherapy, and this information can be used preemptively to guide decisions regarding the use of preoperative chemotherapy for patients with CRLM undergoing curative-intent surgery. Most importantly, it can help guide the use of chemotherapy for cancer patients within this setting, based on desired versus expected concentrations at the tumor site. Given the model that was developed from patients receiving standard preoperative chemotherapy in the context of a real-life setting, theses findings are generalizable to patients receiving systemic chemotherapy for CRCLM with similar characteristics. Furthermore, the purpose of our study was to test the presented concept using surgical specimens so as to have the gold-standard for response to therapy. With findings supporting the association between tumor-site concentration and TRG, the clinical implications can potentially expand to patients treated with systemic therapy only (not surgical patients). Further, these findings can be used as a model to guide future studies incorporating this concept for patients with other tumors–hypovascular ones specifically. Future studies will be needed to validate these applications.

Due to the complexity and myriad factors governing chemotherapeutic drug transport and eventual penetration into tumor tissue, mathematical modeling has evolved into a reliable methodology to understand chemotherapy delivery, thereby providing a framework to quantitatively examine the relative importance of competing factors and to guide clinical use [44,54,55]. Applying a previously developed model of tumor tissue perfusion, here we describe important parameters driving chemotherapy concentration achieved at the tumor site and response to therapy, with a clinical cohort of patients treated with standard 5-FU-based chemotherapy frontline regimens such as FOLFOX. The mathematical model introduced here is reliable, accurate, and is directly suitable for clinical performance. Hence, it represents a developmental blueprint for use with other classes of agents in the field of anti-cancer therapy, including perhaps targeted drugs or immunotherapeutic agents [43,56,57]. We built the model considering biophysical barriers to diffusion and report important findings for the estimation of tumor site chemotherapy. Remarkably, we were able to accurately predict the response to therapy when incorporating the variables derived from individual patient tumors and treatment factors, as predicted in the mathematical model (Figure 5A–D). Indeed, the ability to a priori predict responses to chemotherapy with standard treatment regimens is critical and continues to be a challenge when choosing among various potentially curative treatment strategies for cancer patients. Specifically, for CRLM patients receiving chemotherapy preoperatively, it is essential to be able to predict the response as it may guide the type of chemotherapy regimen to be used and/or may change the strategy to a surgery-first or surgery-only approach. We envision having this model available in the clinical setting at initial patient consultation to help guide individualized patient care by estimating tumor site concentration and predicting expected response to chemotherapy prior to choosing the best treatment approach for a given patient. Similarly, for those patients with metastatic lesions not amenable to surgical approaches, this work can serve as the basis to develop and/or implement novel modes of more effective chemotherapy delivery, such as transarterial infusion (e.g., hepatic artery infusion for CRLM) [58,59], or different strategies for delivering and “dosing” systemic chemotherapy [15,60]. As such, the use of this model will provide more robust information ultimately translating into an individualized, outcome-driven approach to chemotherapy treatment for patients with solid malignancies.

Several aspects of this investigational study merit further comment. Given the retrospective nature of this initial proof-of-concept study, there are some obvious limitations to the data interpretation. Patients included in the analysis may evidently be subject to selection bias favoring those ultimately making it to surgery, and as such exhibiting a different profile in regards to response to chemotherapy. However, we selected serial patients from a real-life surgical oncology practice from a single-institution (an NCI-designated Comprehensive Cancer Center), including all patients receiving preoperative chemotherapy and having resection, as an accurate representation of clinically uniform patients with CRLM being evaluated for surgical treatment in a quaternary setting. Notably, despite the relative similarities of patients included in regard to tumor burden (resectable) and treatment approach (all treated with standard first-line chemotherapy regimens), we still encountered marked variability in the response to treatment and heterogeneity in the tumor tissue perfusion and the overall eTSCC accomplished; this finding resulted in the ideal translational setting to provide an experimental proof-of-concept supporting our working hypothesis regarding tumor site chemotherapy concentration and response to treatment. Similarly, as with mathematical model development, a number of assumptions related to the biophysical barriers of diffusion across to tumor sites were made when developing the model. However, these assumptions are derived from validated work examining each of the different variables encompassed in a real-life model development.

In conclusion, these findings support the working hypothesis that response to chemotherapy is driven by the ability of the chemotherapeutic agent to reach the cancer cells at an effective concentration. Understanding tumor perfusion and chemotherapy delivery principles are essential steps to improve current delivery methods that can be translated into better prediction and improved response to therapy [15,43,51]. This may be particularly relevant when treating hypovascular tumors for which diffusion barriers limit chemotherapy delivery and overall response to treatment. Thus, this proof-of-concept study represents an initial step towards a paradigm shift in our capability to predict the effectiveness of chemotherapeutic agents. If successfully confirmed and reproduced in carefully planned prospective translational studies, here and elsewhere, this work may be considered to be fundamental groundwork for the field in going forward.

## 4. Methods

*Study Design and Population*. A retrospective cohort study was done on patients with CRLM treated with preoperative chemotherapy followed by curative-intent surgery (liver resection-hepatectomy). The study protocol was approved by the Study Review Committee (SRC) at the Moffitt Cancer Center and by the Institutional Review Board (IRB) of the University of South Florida (IRB #MCC16466). The study sample was obtained by query of the hepatobiliary surgical database at Moffitt Cancer Center to identify patients having liver resection for CRLM within the last 30 months (from January 2016 to June 2018), for which tumor tissue specimens were available. Among the initially evaluated patients (*n* = 99), we included only those receiving standard preoperative chemotherapy regimens within four months of surgery, and for whom the oncology record was available for information regarding the chemotherapy regimen and unique actual dose given during each cycle. Lastly, only patients with contrast enhanced pre-chemotherapy computed tomography available for appropriate imaging review of tumor imaging characteristics were included in the analysis, yielding the total cohort of patients reported here (*n* = 33) (Appendix A). Patients included in the study had all the information and data necessary for modeling and analysis. Baseline and treatment characteristics of the study sample are described (Table 1).

*Outcomes*. The aim of this study was to examine the differences in plasma and tumor-site chemotherapy concentration and to evaluate the association of these with response to chemotherapy, which was measured with pathologic TRG, as described in Reference [38]. In short, TRG was expertly assessed by two experienced gastrointestinal pathologists blinded to all other patient and treatment-related information. Archived liver specimens from the hepatectomy procedure were retrieved and slides from the largest lesion were selected and examined for residual tumor cells and the extent of fibrosis. Based on the findings, a TRG score was assigned to each patient using a validated algorithm that incorporates the degree of fibrosis and residual tumor cells; TRG 1: no residual tumor cells or abundant fibrosis; TRG 2: rare residual tumor cells scattered throughout abundant fibrosis; TRG 3: residual tumor cells throughout predominant fibrosis; TRG 4: large amount of tumor cells predominating over fibrosis; and TRG 5: exclusively residual tumor cells without fibrosis [38]. Notably, two pathologists performed the assessment of TRG and they were blind to all the other information. Similarly, the radiologist measuring tumor perfusion variables was blinded to the primary outcome–TRG.

*Data collection*. Patients in the hepatobiliary surgical database are entered prospectively in a consecutive manner, and as such, all surgical cases are included. The database consists of baseline clinical variables (age, gender, race/ethnicity and comorbidities), cancer-related information (primary tumor-site, stage, and metastatic site/s), detailed data regarding the liver metastases, including: presentation, number of lesions, size of largest lesion, serum CEA level at presentation, molecular and mutational analysis (when performed), and treatment characteristics (use of preoperative chemotherapy, margin-negative resection, postoperative complications, and use of adjuvant therapy). Specific variables related to this study, and not collected routinely in the database (such as TRG, chemotherapy regimens and tumor perfusion variables) were collected retrospectively, as described in the corresponding sections.

*Chemotherapy treatment*. For this study, we collected more detailed information regarding the preoperative chemotherapy treatment received by those meeting eligibility criteria. Data collected included chemotherapy agent/s received, BSA-based dose received for each agent during every individual treatment, and number of treatments (cycles). Patients with missing data for any of these variables, those receiving second-line or greater chemotherapy regimens, those treated for >20 cycles or receiving treatment for four months or more before the liver resection procedure were excluded from the analysis.

Tumor perfusion. Based on previous work from our group and a validated method for perfusion assessment of the tumor [13,34,39,43], we measured perfusion of the normal liver and the largest metastatic tumor (to correlate imaging perfusion measurements to the corresponding TRG measurement of the same lesion) with contrast-enhanced three-phase CT (non-contrast, arterial and portal venous phases) obtained prior to the initiation of preoperative chemotherapy of all selected patients. CT measurements were done by two experienced cancer radiologists and a hepatobiliary surgeon, who used Centricity Universal Viewer version 6.0 (GE Healthcare, Waukesha, WI). Manual 1 cm circular regions of interest were applied within the tumor and the surrounding liver on each contrast phase, and the mean Hounsfield units (HU) and standard deviation were recorded. These data were used to calculate tumor perfusion by applying a previously described methodology [13,34,39,43].

Mathematical model development. By analyzing the diffusional transport of drug molecules within the tumor microenvironment, and assuming that local tumor cell death in a tumor is equivalent to the dose response in vitro at the same drug concentration as experienced at that location within the malignant tissue, we obtained a “master equation,” i.e., a closed-form solution of the diffusion equation, predicting the fraction $f_{\mathrm{kill}}$ of a tumor that would be killed by a given chemotherapy regimen:(4)$$f_{\mathrm{kill}}=2\cdot f_{\mathrm{kill}}^{0}\cdot \mathrm{BVF}\cdot \frac{{\mathrm{BVF}}^{1/2}\cdot K_{1}\left(r_{b}/L\right)-K_{1}\left({\mathrm{BVF}}^{-1/2}\cdot r_{b}/L\right)}{{\mathrm{BVF}}^{1/2}\cdot r_{b}/L\cdot K_{0}\left(r_{b}/L\right)\cdot \left(1-\mathrm{BVF}\right)}$$
where *K*_0_ and *K*_1_ are modified Bessel functions of the second kind of orders 0 and 1, respectively; $f_{\mathrm{kill}}^{0}$ is the fraction of kill (dimensionless: $0\leq f_{\mathrm{kill}}^{0}\leq 1$) that would be obtained in a dose-response study in vitro at a drug concentration equal to the time-averaged concentration achieved in the tumor blood vessels throughout the entire multi-month course of the treatment; *L* (units of length) is diffusion penetration distance of drug molecules within tumor tissue, consistent with an observed correlation between lesion size and perfusion (Figure 2D); *r*_b_ (units of length) is the average blood vessel cross-sectional radius, and the BVF (dimensionless) is the blood volume fraction, i.e., the fraction of tumor volume occupied by blood, representing tumor perfusion (0≤BVF≤1). These are measurable patient-specific parameters that characterize the unique pharmacokinetic, as well as the mass transport properties of the patient’s tumor tissue. It can be shown that $f_{\mathrm{kill}}^{0}$ is the maximum theoretical kill achievable in vitro, as transport barriers in patient tumor tissue pose an additional penalty factor (function of BVF and $r_{b}/L$) limiting kill further, thus $f_{\mathrm{kill}}\leq f_{\mathrm{kill}}^{0}\leq 1$. For hypovascular tumors, we can assume BVF≪1, which leads to the simplified form (by 1st order Taylor-series expansion in the BVF in the neighborhood of BVF≈0):(5)$$f_{\mathrm{kill}}\sim f_{\mathrm{kill}}^{0}\cdot \mathrm{BVF}$$
which provides a simple linear dependence on the dose-response $f_{\mathrm{kill}}^{0}$ and the micro-environmental transport factor BVF. Note that there is an additional factor, $\frac{2\cdot K_{1}\left(r_{b}/L\right)}{r_{b}/L\cdot K_{0}\left(r_{b}/L\right)}$, function of the lengths ratio, which is assumed here to be a constant. For example, for a typical vessel radius $r_{b}=10\;\mu m$ and diffusion penetration distance L~100 μm [34], one obtains $r_{b}/L=0.1$ and $\frac{K_{1}\left(r_{b}/L\right)}{r_{b}/L\cdot K_{0}\left(r_{b}/L\right)}=245$. This, as well as other constants, are herein removed from consideration by normalization.

Systemic (plasma) concentration kinetics of drug during one treatment cycle. In a single treatment cycle, the patients received 5-FU as part of the FOLFOX and/or FOLFIRI regimen as an intravenous bolus at a dose D of 400 mg·m^−2^ of BSA followed almost instantaneously, by an intravenous infusion of 5-FU at a dose R of 2400 mg·m^−2^ BSA spread over a duration Δt of 46 h. We began by estimating the blood concentration kinetics of 5-FU with a two-compartment pharmacokinetic modeling approach (Appendix A) [40,41]. The model consists of a system of two ordinary differential equations (ODEs) representing the concentration of 5-FU in the blood compartment ($\sigma_{B}$) and the peripheral compartment ($\sigma_{P}$):(6)$$\frac{d\sigma_{B}}{dt}=-k_{12}\sigma_{B}-\sigma_{B}\frac{\mathrm{Cl}}{V}+k_{21}\sigma_{P}+\frac{D}{\Delta t\cdot V}\cdot \mathrm{BSA},\;\;\sigma_{B}\left(0\right)=\sigma_{0}=\frac{D\cdot \mathrm{BSA}}{V}$$
(7)$$\frac{d\sigma_{P}}{dt}=k_{12}\sigma_{B}-k_{21}\sigma_{P},\;\;\sigma_{P}\left(0\right)=0$$
where $k_{12}$ and $k_{21}$ are the first-order inter-compartment mass transfer rate constants; Cl is the clearance rate of 5-FU; and V is the volume of blood. Parameter values were obtained from published reports and include: $k_{12}=5.35$ h^−1^; $k_{21}$ = 5.69 h^−1^; Cl = 65.3 l·h^−1^ [40,41]. Patient-specific volume of blood (units, mL) was estimated from patient-specific BSA (units, m^2^), by using the following empirical relation [61]:(8)V=(3.29·BSA−1.229)·1000

Assuming concomitant administration of the bolus and infusion doses, we use the theoretical blood concentration $\sigma_{0}$ of 5-FU, achieved instantaneously after the intravenous bolus, as the initial condition for Equation (6). However, in this case, an infusion was not preceded by a bolus injection, the initial condition for Equation (6) was $\sigma_{B}\left(0\right)=0$. We numerically solved the system of equations in MATLAB (MathWorks, Natick, MA) to obtain the blood concentration kinetics curve of 5-FU (Appendix A).

Systemic (plasma) drug concentration kinetics over multiple treatment cycles. From the blood concentration kinetics curve, we determined the area under the curve (${\mathrm{AUC}}_{i}$) (Appendix A) with the trapezoidal method, which is used to obtain the time averaged concentration ${\bar{\sigma }}_{B,i}$ of 5-FU in a given treatment cycle i as:(9)$${\bar{\sigma }}_{B,i}=\frac{{\mathrm{AUC}}_{i}}{\Delta t}$$

By using the PK model to simulate the multiple cycle chemotherapy regimen (Appendix A), we obtained the time averaged blood concentration ${\overset{=}{\sigma }}_{B,i}$ of 5-FU for each patient over N cycles, i.e., the average blood concentration of 5-FU during any given treatment cycle:(10)$${\overset{=}{\sigma }}_{B,i}=\frac{\sum_{i}{\mathrm{AUC}}_{i}}{N\cdot \Delta t}$$

Alternatively, ${\overset{=}{\sigma }}_{B,i}$ can be calculated by averaging ${\bar{\sigma }}_{B,i}$ over N cycles:(11)$${\overset{=}{\sigma }}_{B,i}=\frac{\sum_{i}{\bar{\sigma }}_{B,i}}{N}$$

Classifying the patients based on their response to chemotherapy (TRG), the mean ± SD of ${\overset{=}{\sigma }}_{B,i}$ was plotted as a function of TRG (Figure 4, red squares).

Estimating tumor drug concentration over the course of treatment. In previous work we showed that cumulative uptake of drug over time by cancer cells in vitro, over the course of drug exposure, is a key determinant of drug efficacy [43]. In a clinical setting, this can be equivalently described as the $f_{\mathrm{kill}}$ (tumor cell kill from chemotherapy) achieved through the time averaged drug concentration $\langle \sigma_{T}\rangle$ in the tumor extravascular tissue over the course of treatment t. Here, t represents the time from the initiation of therapy to the time of surgery (Appendix A, inverted blue triangles), thereby also accounting for the intermediary period between cycles and the time after the final cycle, during which drug was not being administered to the patients. Further, based on our own previous work [34] and from earlier discussion of the model in the present report, $f_{\mathrm{kill}}$ correlates with patient-specific tumor perfusion characteristics, i.e., BVF, thus for hypovascular tumors:(12)$$f_{\mathrm{kill}}\sim \langle \sigma_{T}\rangle \sim f_{\mathrm{kill}}^{0}\cdot \mathrm{BVF}$$
where BVF is estimated as:(13)$$\mathrm{BVF}\sim \frac{{\mathrm{AUC}}_{\mathrm{tumor}}}{{\mathrm{AUC}}_{\mathrm{liver}}}$$

${\mathrm{AUC}}_{\mathrm{tumor}}$ and ${\mathrm{AUC}}_{\mathrm{liver}}$ are determined from the contrast enhancement kinetics plots of tumor and liver in an abdominal CT scan, respectively. Clinically, $f_{\mathrm{kill}}^{0}$ is proportionate to the time averaged drug concentration achieved in the tumor vasculature $\langle \sigma_{T,B}\rangle$, which correlates with systemic drug concentration (${\overset{=}{\sigma }}_{B,i}$), frequency of chemotherapy cycles $\left(\lambda =\frac{N}{t}\right)$, and tumor burden (r) as:(14)$$f_{\mathrm{kill}}^{0}\sim \langle \sigma_{T,B}\rangle \sim {\overset{=}{\sigma }}_{B,i}\cdot \frac{\lambda \cdot \Delta t}{r}$$

Given the multiple tumor lesions in every patient, we use a clinically measured blood biomarker of total tumor burden, carcinoembryonic antigen (CEA), as a measure of tumor size, based on the statistically significant correlation observed (Figure 2A). Thus, the effective tumor drug concentration $\langle \sigma_{T}\rangle$ (Figure 4, blue circles) is estimated as:(15)$$\langle \sigma_{T}\rangle \sim {\overset{=}{\sigma }}_{B,i}\cdot \frac{\lambda \cdot \Delta T}{\mathrm{CEA}}\cdot \mathrm{BVF}$$

Binary classification. Logistic regression-based binary classification was performed to test the ability of eTSCC $\langle \sigma_{T}\rangle$ to classify the patients into responders and non-repsonders (Figure 5). The entire data set (*n* = 33) was used to train the binary classifier. A logistic regression model was fit between the predictor ($\langle \sigma_{T}\rangle$) and response (responder/non-responder) variables, and receiver operating characteristic (ROC) curve was computed. Accuracy of classification was obtained as the percentage of tumors correctly classified by the ‘discrimination threshold’ (selected from the ROC curve to maximize the accuracy of classification).

Leave-one-out cross validation (LOOCV) technique was used to evaluate the predictive capability of the binary classifier. In this technique, *n* − 1 training data sets were generated from the total *n* data points by iteratively removing one data point. Each training dataset was used to generate a ROC curve and select a discrimination threshold to classify the left-out test data point. The prediction results from all iterations were pooled to calculate the average accuracy of the classifier.

Statistical analysis. Clinical data are presented as mean ± SD and proportions and range, for continuous and categorical data (at *n* = 33). One-way ANOVA was performed to compare multiple groups and student’s *t*-test was performed for comparison between two groups. A level of *p* < 0.05 was considered statistically significant. The Levenberg-Marquardt algorithm was used to perform regression analysis. All analyses were performed in MATLAB R2018a.

Additionally, multi-variable logistic (ordinal) regression was performed to test the significance of age, gender, tumor presentation (metachronous versus synchronous), lesion count, extra-hepatic disease, and primary tumor location as predictors of therapy response, with the following model to determine the log odds of being in a TRG ≤i versus being in a TRG >i:(16)$$log\left(\frac{P\left(\mathrm{TRG}\leq i\right)}{P\left(\mathrm{TRG}>i\right)}\right)=\alpha_{i}+\beta_{1}\cdot X_{age}+\beta_{2}\cdot X_{gender}+\beta_{3}\cdot X_{presentation}+\beta_{4}\cdot X_{count}+\beta_{5}\cdot X_{EH\;dz}+\beta_{6}\cdot X_{location}$$
where $\alpha_{i}$ is the intercept corresponding to the equation for TRG=i; $\beta_{j}$ represents the regression coefficient of a variable (j=1 for age, j=2 for gender, j=3 for tumor presentation, j=4 for lesion count, j=5 for extrahepatic disease, and j=6 for primary tumor location). Our analysis reveals *p* > 0.05 for all intercepts and coefficients, suggesting that the tested variables cannot reliably predict therapy response. Parameter estimates are given in (Appendix A).

## Figures and Tables

**Figure 1 cancers-13-00444-f001:**
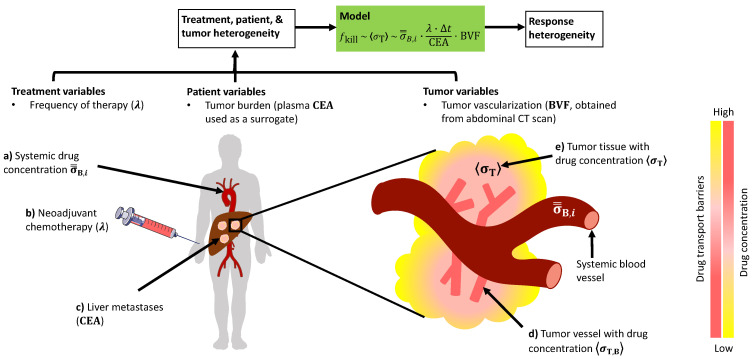
Schematic description of the mathematical model and its variables. The model presented here takes as input the treatment, patient, and tumor-related variables to predict tumor response to chemotherapy. (**a**) Calculated systemic drug concentration, as obtained through pharmacokinetic analysis of patient-specific dosage regimen, is scaled by the (**b**) frequency of drug administration and (**c**) total tumor burden (surrogate: serum carcinoembryonic antigen (CEA)) to provide an estimate for (**d**) tumor vascular drug concentration, which upon further scaling with the tumor blood volume fraction (BVF), estimates drug concentration in the (**e**) tumor interstitium, which correlates directly with response. Note: Normalized area under the curve (AUC) of contrast enhancement kinetics in abdominal CT (computed tomography) scan provides a measure of tumor BVF. Δ*t* is the duration of a single therapy cycle. Color gradients denote level of drug concentration and drug diffusion barriers in the tumor. Illustration is not to scale.

**Figure 2 cancers-13-00444-f002:**
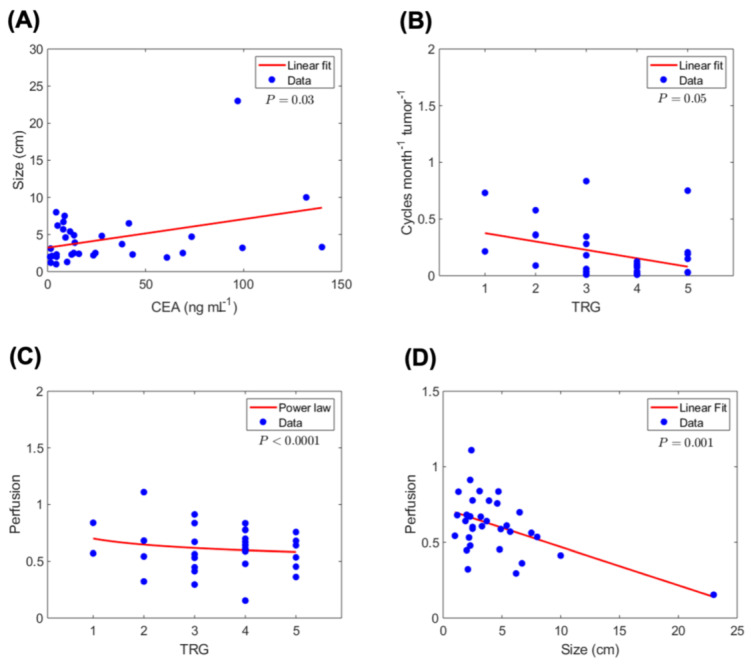
Correlation analysis of tumor, treatment and outcomes variables. Statistically significant correlations are observed between tumor and treatment-related variables and pathological TRG (*n* = 33 patients). (**A**) Correlation of serum CEA with tumor size (largest dimension). (**B**) Correlation of therapy cycle rate per unit tumor burden (i.e., number of treatment cycles per month per unit CEA) with TRG. (**C**) Correlation between tumor perfusion and TRG. This measurement is a surrogate for the blood volume fraction (BVF) parameter in the model. (**D**) Tumor perfusion correlates negatively with tumor size, indicating reduced perfusion in larger lesions.

**Figure 3 cancers-13-00444-f003:**
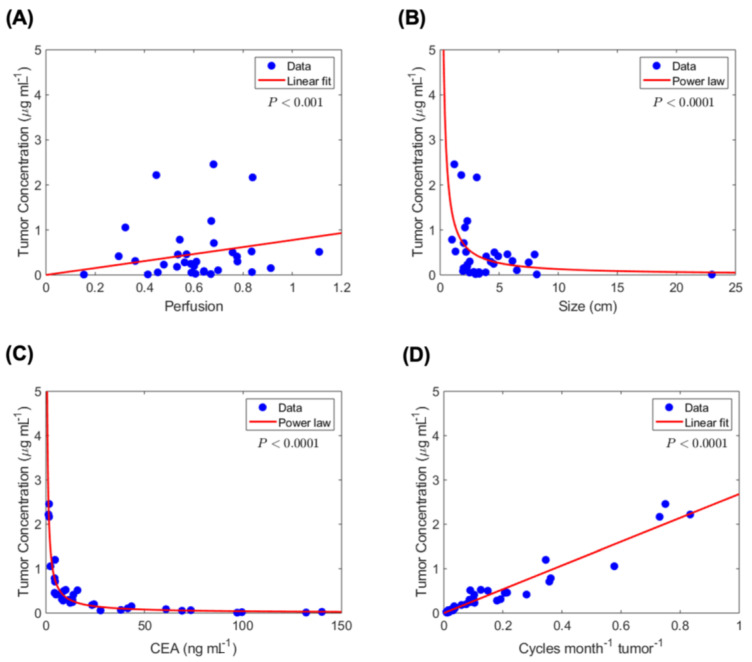
Estimated tumor-site chemotherapy concentration correlation with diffusion-and clinical-based variables. Statistically significant correlations are observed between tumor-site chemotherapy concentration and (**A**) tumor perfusion, (**B**) tumor size, (**C**) serum CEA level, and (**D**) chemotherapy dose overtime per unit tumor burden.

**Figure 4 cancers-13-00444-f004:**
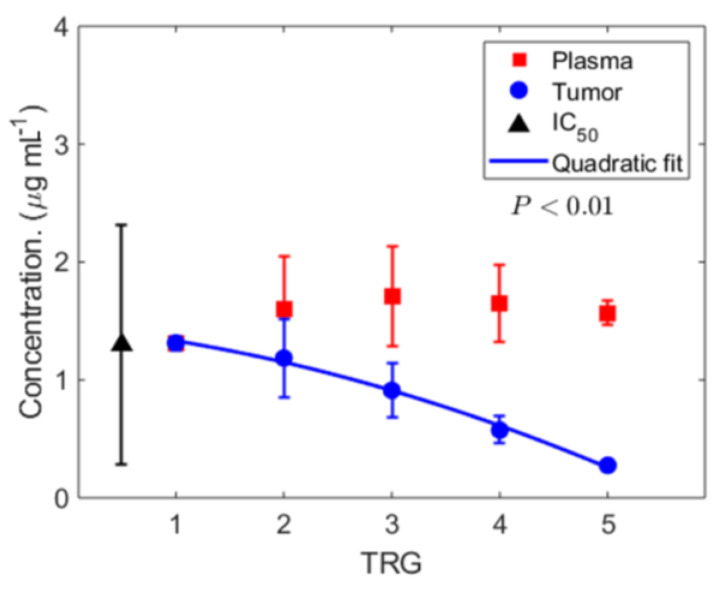
Calculated chemotherapy concentration and response to therapy. Calculated time-averaged plasma concentration ${\overset{=}{\sigma }}_{B,i}$ (red squares) and estimated time-averaged tumor-site concentration of chemotherapy $\langle \sigma_{T}\rangle$ (blue squares with best fit, *p* value < 0.01, $R^{2}=0.98$) throughout the course of multiple cycles of treatment is predicted by PK analysis and the mathematical model. Note that $\langle \sigma_{T}\rangle$ < ${\overset{=}{\sigma }}_{B,i}$, and in particular for TRG = 5, the average tumor concentration is ~5.7 times smaller than the corresponding average plasma concentration, nearly matching the lower bound of IC_50_ for 5-FU. Data represents mean ± standard deviation (SD). (Patient cohort size, *n* = 33).

**Figure 5 cancers-13-00444-f005:**
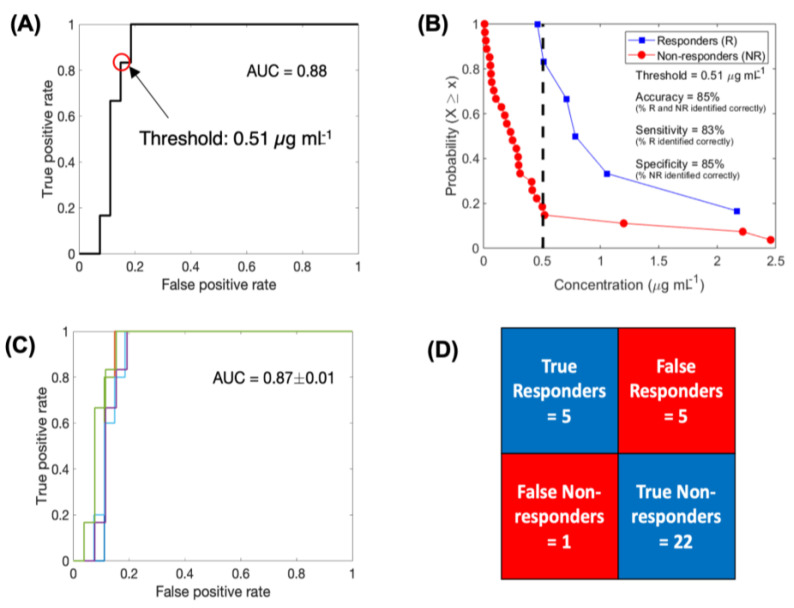
Logistic regression-based binary classification and cross-validation. Receiver operating characteristic (ROC) curve to evaluate the classification ability of eTSCC into responders and non-responders (**A**). Complementary cumulative distribution function (CCD) of patients shows the accuracy of binary classification at a discrimination threshold of 0.51 µg mL^−1^ (**B**). ROC curves generated for multiple training data sets obtained through the leave-one-out cross validation technique (**C**). Results of cross validation in correctly classifying the test data point (**D**).

**Table 1 cancers-13-00444-t001:** Features of patients with colorectal liver metastasis, followed by partial liver resection after chemotherapy.

Clinical Characteristics	All (*n* = 33)
**Patient features**	
Median age	57 (range, 42–83)
Male	20 (60%)
Female	13 (40%)
**Clinical tumor-related characteristics**	
Primary tumor site	
Right colon	12 (36%)
Left colon	12 (36%)
Rectum	9 (28%)
Synchronous presentation–CRLM	29 (89%)
Extrahepatic disease present	11 (33%)
**Colorectal cancer liver metastases**	
Number of lesions (median)	2 (range, 1–5)
Largest lesion size (median)	3.2 cm (range, 1–10)
CEA at presentation (median)	13.4 (range, 1–97)
Molecular characteristics	
MSI-high tumors	0
Kras mutated tumors	11 (33%)
BRAF mutated tumors	1 (3%)
**Treatment variables**	
Median of preoperative chemotherapy cycles	4 (range, 1–9)
Chemotherapy regimen	
FOLFOX	25 (76%)
FOLFIRI	1 (3%)
FOLFOX/FOLFIRI combinations	7 (21%)
Combination with targeted/biologic agent	
None	13 (39%)
Bevacizumab	20 (61%)
**TRG outcomes**	
TRG1	2 (6%)
TRG2	4 (12%)
TRG3	8 (24%)
TRG4	13 (39%)
TRG5	6 (19%)

## Data Availability

Data is available from the authors upon request.

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
