# Peer review of "A Mathematical Model to Estimate Chemotherapy Concentration at the Tumor-Site and Predict Therapy Response in Colorectal Cancer Patients with Liver Metastases"

_cancers, 2021, doi:10.3390/cancers13030444_

Round 1

Reviewer 1 Report

The authors have made adequate changes to the manuscript to meet my prior concerns and I now recommend acceptance.

Reviewer 2 Report

I'm satisfied with the changes done by the authors. Maybe a graphical abstract might improve the clearness of the results.

This manuscript is a resubmission of an earlier submission. The following is a list of the peer review reports and author responses from that submission.

Round 1

Reviewer 1 Report

Thank you for the opportunity to revise this manuscript. The topic is really interesting and relevant. Some aspects should be addressed before publication. 

Authors should explain how the study size was arrived at 33 patients. The flow diagram of included/excluded participants should be presented with reasons of exclusion. In particular, indicate number of participants with missing data for each variable of interest.

Authors should be more informative regarding the blindness. clearly state who was blinded and when. 

Authors stated that this is a retrospective data, but in data collection section they stated "prospective data collection". Could you clarify this aspect? What do they mean with prospective data collection in a retrospective study?

Authors should be more detailed in the definition of the TRG score. As for instance what does "abundant fibrosis" mean? what "rare residual", or "predominant" and so on. Please provide cut-off.

Authors should discuss the generalisability of the study results

Reviewer 2 Report

Authors develop a mathematical model to estimate drug concentration at the tumor and relate this to outcomes. It is reasonable to expect therapy response is related to the amount of drug that reach the tumor site, in this case liver metastases, and the authors develop a mathematical model to test if this is true. The investigators quantify tumor blood volume fraction (BVF) from contrast-enhanced CT scan. Total tumor burden is estimated from serum CEA values. A mathematical model is developed relating these measurable parameters to the expected fraction of tumor cells kills based on principles of diffusion. The paper is in general well written and based on careful analysis of clinical data.

Small lesions will respond better because of greater perfusion but they may respond better because they are less aggressive or because they were detected earlier. So this factor could be confounded by factors not related to drug delivery. Further, how the length (L) parameter is quantified is not described

Surprising that the model is useful when extrahepatic disease is present since none of the measurements or even the model itself is related to extra-hepatic disease. This should be commented on and justification for including 33% of patients with extrahepatic disease should be stated.
